

# Functional responses of male and female European green crabs suggest potential sex-specific impacts of invasion

Kiara R. Kattler, Elizabeth M. Oishi, Em G. Lim, Hannah V. Watkins and Isabelle M. Côté

Biological Sciences, Simon Fraser University, Burnaby, BC, Canada

Corresponding author
Isabelle M. Côté, imcote@sfu.ca

## ABSTRACT

Predicting the impacts of predatory invasive species is important for prioritising conservation interventions. Functional response experiments, which examine consumption by predators in relation to prey density, are a useful way to assess the potential strength of novel predator-prey relationships. However, such experiments are often conducted without consideration of sex or only with males to reduce invasion risk. Here, we compared the functional responses of male and female European green crabs (*Carcinus maenas*), a global invader, feeding on varnish clams (*Nuttallia obscurata*) to test whether the two sexes have similar potential for impact. We also examined potential correlates of predation behaviour by measuring sex-specific movement and prey choice. Both sexes displayed a Type II hyperbolic functional response, which can destabilise prey populations at low prey densities. However, males and females exhibited some differences in foraging behaviour. Female green crabs had slightly lower attack rates, which were not linked to sex differences in movement, and slightly longer handling times, which were not linked to sex differences in prey choice. These small, non-significant differences nevertheless translated into significantly greater functional response ratios, which are used to predict the ecological impact of invasive species, for males than females. There was no difference in the proportion of clams consumed between males and females with similar crusher claw heights, but females have smaller crusher claws on average, hence they consumed a smaller proportion of clams. Repeated surveys of four populations of European green crabs established in British Columbia, Canada, showed that sex ratio is highly variable. Taken together, these results and population-level modelling suggest that trying to evaluate the potential impact of European green crabs on clam populations by sampling only males could result in overestimation, even in populations that have male-biased sex-ratios. Consumer sex might generally be an important feature to consider when using functional response experiments to forecast the impact of new invasive species, especially those with marked sexual dimorphism that affect foraging.

## INTRODUCTION

Invasive species have been altering ecosystems around the world for centuries, with little sign of abating (*Seebens et al., 2017*). Marine habitats are no exception (*Molnar et al.,*

*2008*), and introductions through commercial shipping (*e.g.,* ballast water, hull fouling) and marine aquaculture (*e.g.,* intentional release for stock enhancement, discarded traps, live packing materials) are primary causes of marine invasions (*Bax et al., 2003*; *Williams et al., 2013*). Marine invaders often have detrimental impacts on native species, community biodiversity, and ecosystem services (*Grosholz, 2002*; *Katsanevakis et al., 2014*; *Pyšek et al., 2020*), and cause the loss of billions of dollars annually to coastal economies (*Cuthbert et al., 2021*). As a result, predicting impacts before introductions and monitoring for early detection have become critical management tools.

A key determinant of the impact of invading consumers is their potential for resource use. A commonly used method to assess this potential is functional response experiments, which examine the relationship between the consumption rate of a predator as a function of the density of prey (*Holling, 1959*). The shape of functional responses provides essential quantitative information on various predation processes, namely attack rate, handling time, and maximum prey consumption (*Holling, 1959*; *Alexander et al., 2012*). It also reveals the potential for predators to destabilise prey populations, particularly at low prey densities when predators can find and consume all prey available (*i.e.,* a Type II functional response) (*Holling, 1959*; *Hassell, 1978*). Functional responses can differ between invasive and native species, among invasive species, and within species under different contexts (*Haddaway et al., 2012*; *Dick et al., 2013*; *Alexander et al., 2014*; *Howard et al., 2018*; *Howard et al., 2022*; *Ens et al., 2021*; *Chucholl & Chucholl, 2021*).

While the effects of external abiotic factors (such as temperature, salinity, habitat complexity, light regimes) on functional responses have been previously examined (*e.g., South et al., 2017*; *Howard et al., 2022*; *Cuthbert & Briski, 2022*), there is usually little consideration of consumer sex. Functional response experiments often pool sexes together (*e.g., Usio et al., 2009*), or do not specify which sex is used (*e.g., Alexander et al., 2014*; *South et al., 2017*). Those that target invasive predators often rely on the study of males (*e.g., Howard et al., 2018*; *Howard et al., 2022*; *Ens et al., 2021*; *Chucholl & Chucholl, 2021*). There is a good reason for this bias: the risk of accidental introductions is eliminated if females, especially gravid ones, are not transported to experimental facilities. Yet, sex could have an important influence on consumption patterns. Compared to males, females often require more energy intake to fuel egg production (*Trivers, 1972*). Moreover, sexual body dimorphism, when present, can influence energetic requirements and ability to consume prey (*Hayward & Gillooly, 2011*).

In this study, we examine sex differences in the functional responses of a global invader: the European green crab (*Carcinus maenas,* Linnaeus, 1758). Native to Europe and North Africa, European green crabs have become established in intertidal and shallow subtidal zones on the east and west coasts of North America, and throughout South Africa, Australia, South America, and Asia *via* human vectors (*Klassen & Locke, 2007*; *Young & Elliott, 2020*). Their invasion success is partly due to their tolerance of a broad range of environmental conditions (*Behrens Yamada, 2001*; *Young & Elliott, 2020*) and ability to consume a wide variety of prey (*Cohen, Carlton & Fountain, 1995*); (*Klassen & Locke, 2007*; *Young & Elliott, 2020*). European green crabs can alter fish community structure (*Matheson et al., 2016*), compete with native crustaceans (*McDonald, Jensen & Armstrong,*

 

*2001*), degrade eelgrass habitat (*Matheson et al., 2016*; *Howard et al., 2019*; *Young & Elliott, 2020*), and cause shellfisheries declines (*Grosholz et al., 2011*; *Mach & Chan, 2014*). Due to the significant impacts of invasive green crabs on ecosystems, it is crucial to understand how males and females behave and use resources in the habitats they invade. Evidence for sex-linked foraging differences in European green crabs is currently mixed. *Elner (1980)* found that female green crabs prefer smaller mussels and eat more of them each day than males in the native range, while males have broader diets than females in South America (*Cordone et al., 2022*). However, male and female green crabs have also previously been considered ecologically equivalent in their native range given a lack of sex-based foraging niche differentiation (*Baeta et al., 2006*; *Spooner, Coleman & Attrill, 2007*).

Here, we compare the type of functional responses exhibited by female and male green crabs, as well as their attack rates, handling times and proportion of prey consumed by each sex. To explore the potential drivers of varying consumer behaviour, we examine sexual dimorphism in carapace width, crusher claw height, prey choice, and exploratory behaviour. Finally, we scale up our per-capita sex-specific estimates of consumption rates to population levels to examine the potential extent of error in consumption efficiency that arises from considering only males.

## MATERIALS & METHODS

### Functional response experiments
#### *Animal collection*
We collected European green crabs from Bedwell Bay (49.30919, −125.80489) on the west coast of Vancouver Island, British Columbia (BC), Canada, in June 2022 with the Coastal Restoration Society. Males and non-gravid intermoult females, with a notch-to-notch carapace width of 50–76 mm, free from epibionts and with two intact claws, were collected and transported to the Bamfield Marine Science Centre (BMSC), BC.

On the Pacific coast of North America, European green crabs are generalist, omnivorous predators that consume predominantly mollusks and crustaceans (*Jamieson et al., 1998*; *Klassen & Locke, 2007*). As prey species, we therefore used one of the commonest clam species in the region, *i.e.,* the varnish clam (*Nuttallia obscurata*), which co-occurs with green crabs in many areas (S Dudas, pers. comm., 2022). Although these clams are not native to BC, their abundance and physical characteristics (*i.e.,* thin shells) make them a preferred prey species for local green crabs (*Molnar et al., 2008*; *Curtis et al., 2012*). It is important to note that our results, which are based on a single prey species, could be unrepresentative of predation on other types of prey, especially when predation is not affected by physical differences between male and female crabs. Undamaged varnish clams of 25–40 mm in length (*i.e.,* anterior to posterior edge) were collected from Robbers Passage, Barkley Sound, BC (48.89622, −125.12081).

Both crabs and clams were housed separately in indoor sea-tables supplied by flow-through unfiltered seawater at the BMSC. Female and male crabs were kept separately (∼20 crabs per sea-table) and provided with shelter, rocks, and seaweed. Female green crabs were inspected daily to ensure they did not develop eggs, to prevent introduction

into Bamfield Inlet. Crabs were fed every 3–4 days with salmon and clams were fed every three days with concentrated phytoplankton (PhytoFeast). Both species were held under artificial lighting that mimicked the natural day–night cycle.

*Experimental set-up*

We conducted functional response experiments in June and July 2022. Prior to the start of a trial, we selected up to 12 crabs haphazardly from the holding sea-tables, isolated them, and withheld food for 48 h to standardise hunger levels (*Howard et al., 2018*; *Howard et al., 2022*). Trials with female crabs were run first, followed by trials with males to minimise the time female crabs were kept in captivity and thus the risk of accidental introduction of green crabs into Bamfield Inlet. Female green crabs ($n = 36$) were held for 5 to 9 days, and male green crabs ($n = 36$) were held for 15 to 19 days, both including starvation time. Female carapace width ranged from 50.0–76.0 mm (mean $\pm$ SD, 64.9 mm $\pm$ 5.6 mm) and males ranged from 56.0–75.0 mm (67.1 mm $\pm$ 5.0 mm). Crusher claw height for females ranged from 8.0–18.0 mm (13.8 mm $\pm$ 2.4 mm) and 12.0–25.0 mm (17.4 mm $\pm$ 3.1 mm) for males. Opaque plastic enclosures (68.1 L, $61 \times 41 \times 42$ cm) were used for individual trials and each was supplied with an independent source of constant flowing seawater and artificial lighting. During the course of the experiment, we stopped foot traffic around the enclosures to prevent disruptions to crab foraging.

We randomly selected varnish clams from the holding sea-tables, measuring shell length and height (umbo to mouth) before placing them in the enclosures at one of six densities (1, 2, 4, 6, 10, or 16 clams per enclosure) 12 h prior to the beginning of trials. Each density by sex combination was replicated six times ($n = 72$). A control replicate with clams but without a crab was run with one of each of the six densities. We recorded the sex, carapace width and crusher claw height of each starved crab before introducing it to a haphazardly chosen enclosure. Crabs were allowed to feed on the clams for 8 h, starting at 08:00. At the end of a trial, crabs were removed from the enclosures, and the number and size (height and length) of clams remaining were recorded. By comparing post-trial sizes to the relevant pre-trial sizes, we were able to determine the number and sizes of clams consumed in each trial. We conducted up to 12 trials per day, which ran simultaneously. Crabs were only used once and euthanized by freezing at the end of each trial following BMSC animal care protocols. Individual clams were used only once, even if they were not consumed. Temperature and salinity were measured at the start and end of each 8-h trial using a thermometer (Fisherbrand© 76 mm immersion thermometer) and refractometer (Tropic Eden© PRO-1 normal seawater refractometer), respectively. Seawater temperature remained constant throughout the experimental period (10 °C $\pm$ 0.33 °C) and salinity ranged from 33–37 ppt.

## Potential correlates of green crab predation
### Carapace and claw dimorphism

To assess sexual dimorphism, European green crabs were opportunistically collected and measured in August 2022 from the Tranquil River Estuary (49.207, −125.671), ∼35 km from Bedwell Bay, with help from the Coastal Restoration Society. Crabs were collected from 40 traps that were deployed for ∼24 h; of the 2,184 crabs caught, we measured 372

chosen haphazardly. Each crab was sexed and its carapace width and crusher claw height measured using calipers in the field. Forty of the crabs ($n = 20$ males, $n = 20$ females) were brought back alive to BMSC for further behavioural experiments (see below); the rest were euthanized as part of the South Coast European Green Crab Control Project.

### Exploratory behaviour

We examined sex differences in exploratory behaviour by observing crabs as they moved around enclosures. Male and non-gravid female green crabs with both claws intact, a notch-to-notch carapace width between 55–75 mm, and no evidence of moulting or epibionts were collected from the Tranquil River Estuary (see above) and transported to BMSC. Crabs were housed in conditions identical to those of the functional response experiment and were starved for 48 h prior to the behavioural observations.

Exploratory behaviour experiments ran from 08:00 to noon. We suspended one GoPro Hero3 camera 40 cm above each of three enclosures (68.1 L, $61 \times 41 \times 42$ cm), which had no substrate, cover, or flowing water. The enclosures were placed outdoors in the shade to record a clear video without glare from artificial lighting or moving water. The sex, carapace width, and crusher claw height of each crab were recorded. A dot was drawn with a permanent marker on the centre of each crab's carapace as a reference point for tracking movement. Video recording started immediately after a crab was placed in each enclosure and three trials were run at once. We recorded 5-min videos of 16 male crabs and 16 female crabs.

For each crab, we calculated path length, time spent moving, and average moving speed. Path length was analysed using the software Tracker (V 6.0.9, 2022) by placing a tracking point on the focal crab's carapace every 35 frames. The time spent moving was measured with a stopwatch and converted to proportion of overall observation time. Average moving speed was calculated as the path length divided by time spent moving. Because of technical issues with video recording, the sample size for females is variable ($n = 12$ for path length, $n = 13$ for proportion of time spent moving, $n = 11$ for average speed).

### Population sex ratios

We estimated sex ratios of European green crab populations using data collected as part of the South Coast European Green Crab Control Project. The Coastal Restoration Society and Ahousaht and Tla-o-qui-aht Nations repeatedly trapped green crabs at four invaded sites on the west coast of Vancouver Island (Lemmens, Tranquil, Bedwell, Cypre) between November 2021 and October 2022. The crabs were collected with prawn traps baited with herring and soaked for ~24 h. Trapping at each site is typically conducted monthly, with the Coastal Restoration Society rotating among sites and conducting depletion sampling for up to five consecutive days. All crabs caught are counted, sexed, and removed from the population. For this study, the cumulative sex ratio of each trapping period at a site was used, yielding 42 estimates of sex ratio across the four populations.

## Analyses

### Functional response

We assessed functional response curves using a logistic regression that fit the proportion of prey eaten to prey density with the R package "frair" (*Pritchard et al., 2017*). Both male and female green crabs exhibited a Type II functional response (see Results). We therefore fit the data using the Rogers' Type II model equation $N_e = N_0(1 - e^{(a(N_e h - T))})$, which is the random predator equation without prey replacement (*Rogers, 1972*), where $N_e$ is the number of prey consumed, $N_0$ is the initial prey density, $a$ is the attack rate, $h$ is the handling time, and $T$ is the experimental duration (fixed to 1 for model fitting). Using AIC values corrected for small sample sizes and friar::frair_fit with maximum likelihood estimation, we confirmed that the Type II fit was best. To generate 95% confidence intervals of model parameters and to test the differences between male and female green crabs at the highest experimental density of clams (*i.e.*, $N_0 = 16$), we used frair:frair_boot nonparametric stratified bootstrapping ($n = 2,000$ iterations). For each iteration of the bootstrap, we took the difference between the estimated $N_e$ for males and for females at $N_0 = 16$, then examined the resulting distribution of differences. The 2.5% and 97.5% quantiles of this distribution represent the lower and upper bounds of the 95% confidence interval around the difference. If the confidence interval includes 0, the difference is considered non-significant.

We calculated the functional response ratio (FRR, *i.e.*, attack rate/handling time) for each sex for each iteration of the bootstrap. The functional response ratio aims to integrate the various metrics of functional response experiments to predict a species' ecological impacts, which is useful when dealing with invading species (*Cuthbert et al., 2019*). To determine if the FRR varied significantly between the sexes, we calculated $FRR_{male}:FRR_{female}$ for each iteration and then examined the resulting distribution of ratios. The 2.5% and 97.5% quantiles of this distribution represent the lower and upper bounds of the 95% confidence interval around the ratio. If the confidence interval includes 0, the difference between males and females is considered non-significant.

### Predator morphology

We examined the effect of crab size on predation in the functional response experiment using a generalised linear model with a binomial distribution and logit link function, with the proportion of clams eaten as a response variable, and initial prey density, sex, crusher claw height, and a sex:claw height interaction as fixed effects. We also separately ran the same model with carapace width instead of crusher claw height, as the two predictors were too correlated to be included in the same model. We then ran linear models to test for differences in mean carapace width and crusher claw height between male and female crabs in crabs from our experimental trials and from the greater population survey at Tranquil River Estuary.

### Exploratory behaviour

We constructed linear models to examine differences in path length, proportion of time moving, and speed between the sexes, with carapace width included as a fixed effect and an interaction between sex and carapace width.

### Modelling population-level estimates of consumption efficiency

To examine the potential bias in estimation of consumption efficiency (*i.e.,* proportion of clams eaten) when considering only male crabs relative to varying proportions of female crabs in a population, we used the generalised linear model described in the *Predator morphology* section and bootstrapped estimates of male and female consumption efficiency to generate confidence intervals around the proportional differences. For each iteration of the bootstrap, we examined the model predictions for the proportion of clams consumed by an average sized female (*i.e.,* based on the mean crusher claw height observed at Tranquil River Estuary; 11.48 mm, 95% CI: 11.15 mm–11.81 mm, $n = 197$) and an average sized male (mean crusher claw height: 16.71, 95% CI: 16.19 mm–17.23 mm, $n = 175$). We then calculated the proportional difference between each of these estimates and the original model predictions for an average sized male (*e.g.,* $\frac{prop\ clams\ consumed_{female\ crab,bootstrap\ iteration\ 1} - prop\ clams\ consumed_{male\ crab,original\ model}}{prop\ clams\ consumed_{male\ crab,original\ model}}$). We then simulated a series of hypothetical populations ranging in sex ratios from entirely male to entirely female. For each iteration of the bootstrap, we calculated the difference between an all-male population using the original model, and the hypothetical mixed-sex population using the bootstrapped iteration of the model. We took the 2.5% and 97.5% quantiles of the resulting distribution as the lower and upper bounds of the 95% confidence interval around these estimates.

For all analyses, we used R (Version 4.1.2; *R Core Team, 2020*). For all (generalised) linear models presented in this paper, model fit was assessed using the 'DHARMa' package (*Hartig, 2022*), which includes tests for assessing the distribution of residuals, dispersion, outliers, and homogeneity of variance.

## RESULTS

### Functional responses of male and female green crabs

There was no varnish clam mortality in the enclosures without green crabs, suggesting that all mortality in trials with green crabs was due to predation. Both male and female green crabs exhibited a Type II hyperbolic functional response when fed varnish clams, indicated by significant negative first-order density terms in the logistic regression models (females: z = −4.89, $p < 0.01$, males: z = −4.57, $p < 0.01$). The functional response curve of males consistently predicted higher consumption rates than that of females, but there was overlap in the confidence intervals across all prey densities (Fig. 1), and no significant difference in maximum consumption rates (Fig. 2) or the consumption rate at the maximum density of clams used in this experiment (bootstrapped 95% CI on difference: −0.48:3.81, permutation test *p*-value = 0.17; Fig. S1).

The coefficient estimates of attack rate and handling time derived from the functional response models were significant for both males and females (all $p < 0.001$; Table S1). Males had higher attack rates (Fig. 2, mean bootstrapped difference = 0.99, 95% CI = −1.21: 3.86) and maximum consumption rates ($1/hT$, mean bootstrapped difference = 2.18, 95% CI = −2.59: 8.50), as well as lower handling times (mean bootstrapped difference = −0.03, 95% CI = −0.12: 0.04) compared to females. However, in all cases, these differences were not statistically significant.
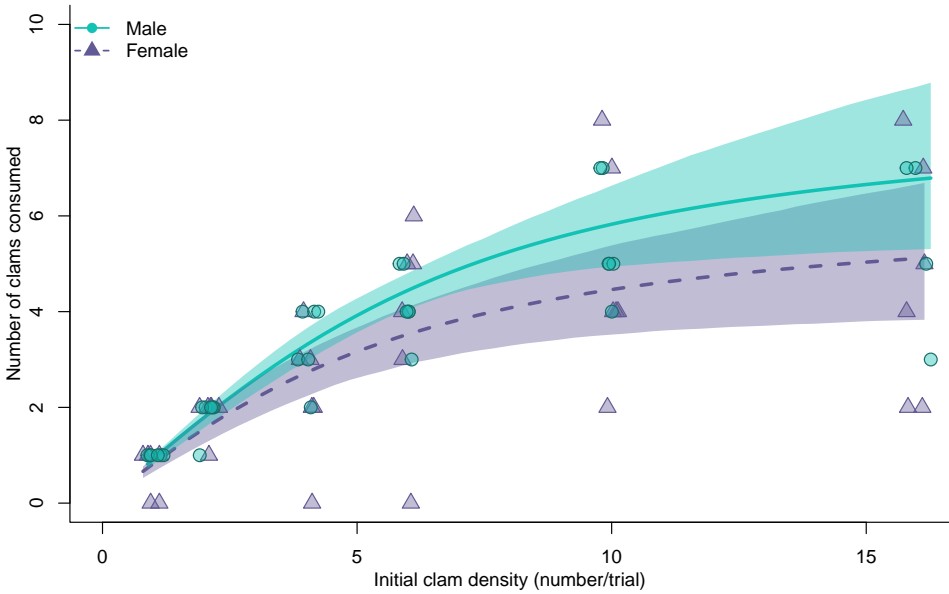

**Figure 1** **Functional response curves for male (blue dots, solid line) and female (purple triangles, dashed line) European green crabs (*Carcinus maenas*), relating the mean number of varnish clams consumed to varnish clam (*Nuttallia obscurata*) density.** Both curves are a Type II response. The shaded areas are 95% bootstrapped confidence intervals.

The estimated functional response ratio was significantly higher for males (mean FRR: 26.05, 95% bootstrapped CI = 18.91: 36.76) than for females (mean FRR:13.74, 95% bootstrapped CI = 8.30: 21.29) (mean FRR difference between sexes = 2.01, 95% CI = 1.09: 3.50).

## Potential correlates of sex differences in prey consumption
### Predator morphology
The proportion of clams consumed by green crabs decreased with increasing prey density (coefficient = −0.76, $p < 0.001$). Female green crab consumption rates increased with increasing crusher claw height (coefficient = 0.59, $p = 0.006$), while crusher claw height had no effect on male green crab consumption rates (interaction between sex and crusher claw height = −0.57, $p = 0.04$; Fig. 3). There was no effect of green crab sex on the proportion of clams consumed at mean crusher claw height (coefficient = 0.29, $p = 0.24$). The results were qualitatively similar when we considered carapace width instead of crusher claw height, although in that case, there was a significant effect of sex (coefficient = 0.46, $p = 0.03$), with males consuming a greater proportion of clams than females at the mean carapace width, but no interaction between carapace width and sex (coefficient = −0.32, $p = 0.16$) (Fig. S2).

For the green crabs used in the functional response experiment, there was no significant difference in carapace width between the two sexes, although male green crabs tended to be slightly larger (2.22 mm, or 3.4%, $p = 0.078$) than females. This is not surprising since we selected crabs from a limited size range. However, male green crabs had significantly larger

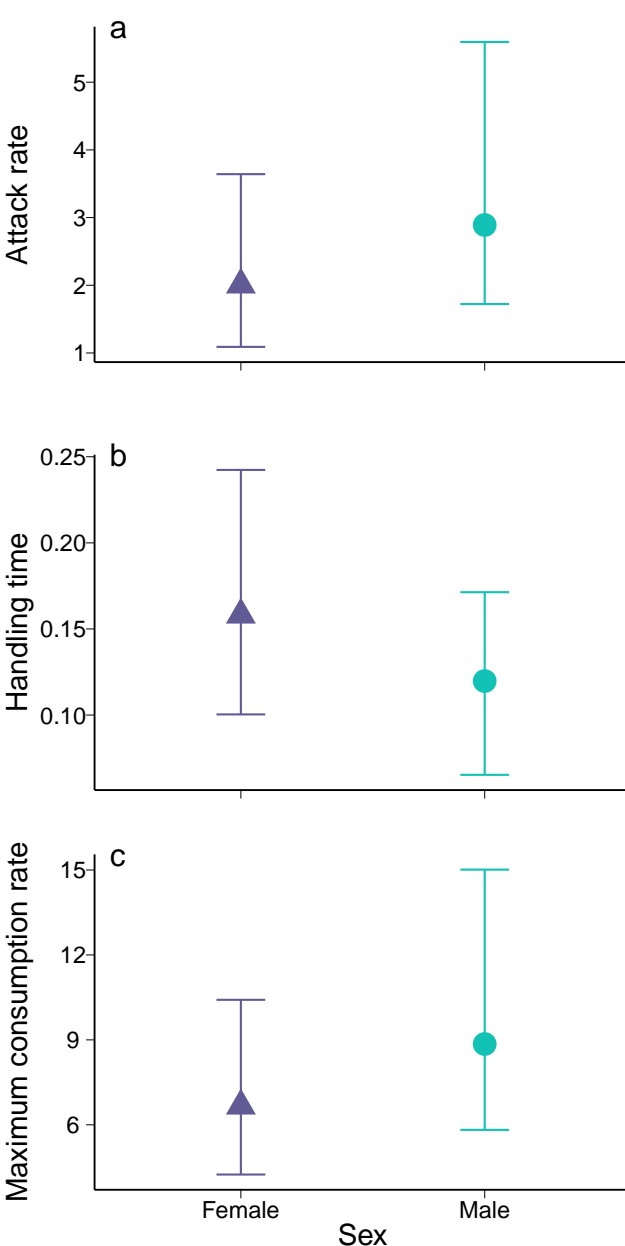

**Figure 2** **Parameter estimates for (A) attack rate, (B) handling time, and (C) maximum consumption rate of female and male European green crabs (*Carcinus maenas*) foraging on varnish clams (*Nuttallia obscurata*).** The parameters were derived from the functional response curves in Fig. 1. Dots and whiskers represent mean estimates and 95% bootstrapped confidence intervals.

crusher claws (3.58 mm, or 25.9%, $p < 0.001$). The differences in carapace and crusher claw sizes observed in our experiment were magnified in the larger sample (372 green crabs: 197 females and 175 males) with unrestricted size range from the Tranquil River. Male carapace was, on average, 16% larger (9.26 mm, $p < 0.001$), and crusher claws 45% larger (5.22 mm, $p < 0.001$) than those of females.

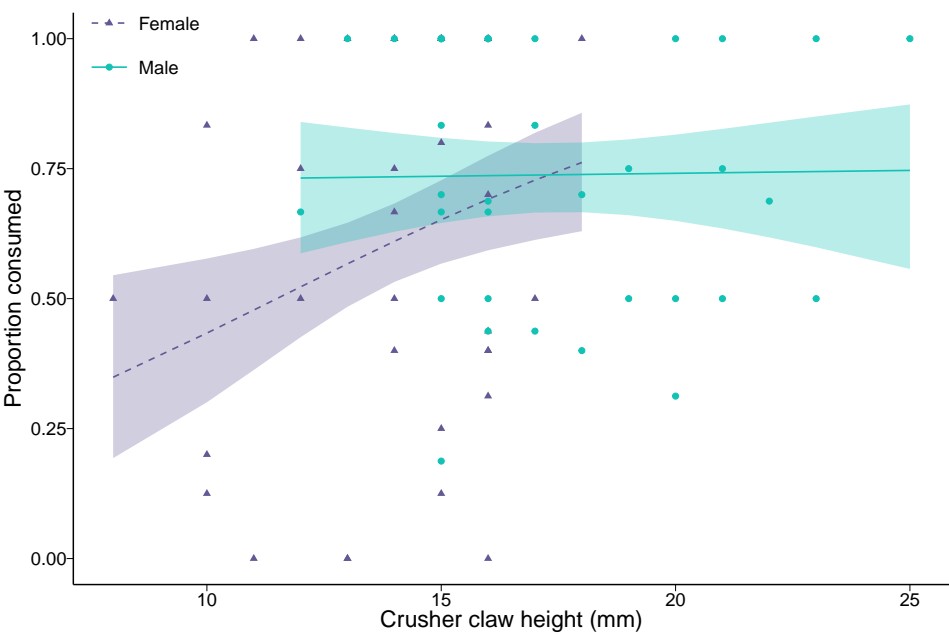

**Figure 3** **Proportion of varnish clams (*Nuttallia obscurata*) consumed in relation to crusher claw height of male (blue dots, solid line) and female (purple triangles, dashed line) European green crabs (*Carcinus maenas*).** Points are observed data; lines are model fit with shaded 95% confidence intervals.

### *Prey morphology*

At the beginning of the trials, males were offered slightly smaller clams on average compared to female crabs (*i.e.,* 2.85 mm or 8.5%) as a result of our random assignment of prey (post-hoc pairwise contrast $p < 0.0001$; Fig. 4). Neither male nor female crabs preferentially consumed smaller or larger prey than they were offered (post-hoc pairwise contrasts $p = 0.17$ for females, $p = 0.72$ for males; Fig. 4).

### *Predator exploratory behaviour*

There were no detectable differences in exploration behaviour between male and female green crabs. Path length did not vary between the sexes (coefficient $= 760.38$, $p = 0.74$; Fig. 5A) or with carapace width (coefficient $= -12.74$, $p = 0.66$), and there was no interaction between sex and carapace width (coefficient $= -8.51$, $p = 0.81$). Male and female green crabs were in movement for a similar proportion of time (coefficient $= -0.38$, $p = 0.81$; Fig. 5B), with no effect of carapace width (coefficient $= -0.03$, $p = 0.21$) or interaction between predictors (coefficient $= 0.01$, $p = 0.71$). Similarly, there was no difference in speed between female and male green crabs (coefficient $= 1.41$, $p = 0.84$; Fig. 5C), as well as no effect of carapace width (coefficient $= 0.03$, $p = 0.75$) or interaction between sex and carapace width (coefficient $= -0.02$, $p = 0.83$). The results were similar when we considered crusher claw height instead of carapace width.

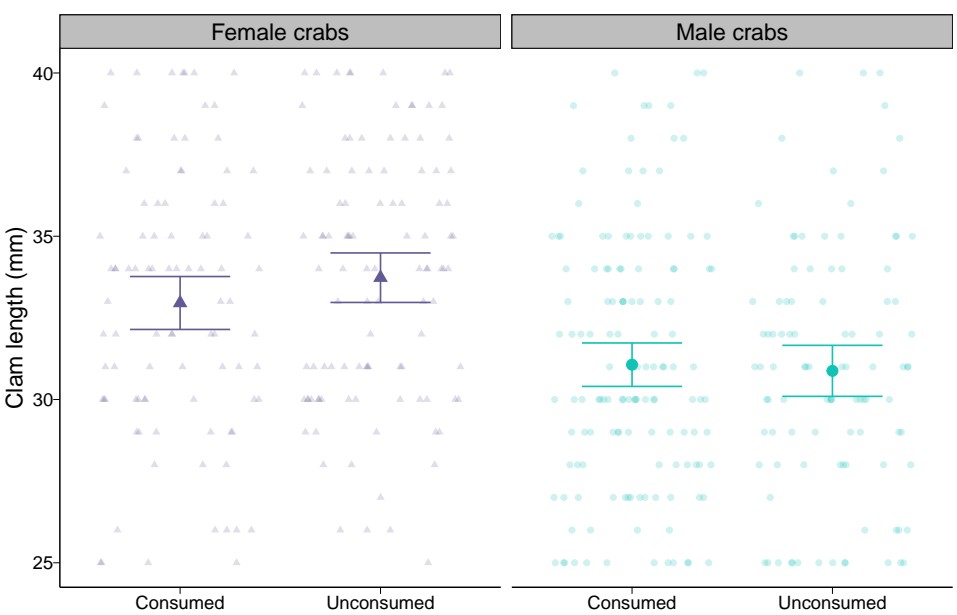

**Figure 4 Length of varnish clams (*Nuttallia obscurata*) consumed or not consumed by female and male European green crabs (*Carcinus maenas*) during foraging trials.** Dots and whiskers show model-predicted mean lengths and 95% confidence intervals. Observed data are shown as smaller points.

## Sex ratio variation in wild populations

The mean proportion of females in each of the four BC green crab populations examined ranged from 0.41 to 0.61, with individual traps ranging between 0 and 1. On average, the populations from Bedwell and Cypre had an even sex ratio, but those of Tranquil and Lemmen were slightly but significantly female- and male-biased, respectively (Fig. S3).

The predicted difference in proportion of clams eaten relative to a male-only population became more negative as the relative number of females in a population increased, indicating an increasing overestimate of potential impact of invasive crabs (Fig. 6). Overestimation of impact becomes nearly certain (*i.e.,* the confidence interval no longer overlaps zero) when populations are at least 31% female. Approximately 90% (38 of 42) of the observed samples had a greater proportion of females than this.

## DISCUSSION

Male and female European green crabs readily consumed varnish clams. Both exhibited a Type II hyperbolic functional response, which suggests a destabilising relationship with prey since predation pressure remains high even at low prey densities. However, male and female green crabs exhibited some differences in foraging behaviour. Females had slightly lower attack rates, slightly longer handling times, and lower maximum consumption rates than males. Although these differences in individual predation parameters were not statistically significant, they translated into a significantly higher functional response ratio for males compared to females. There was no difference in the proportion of clams consumed between males and females with similar crusher claw heights, but females have

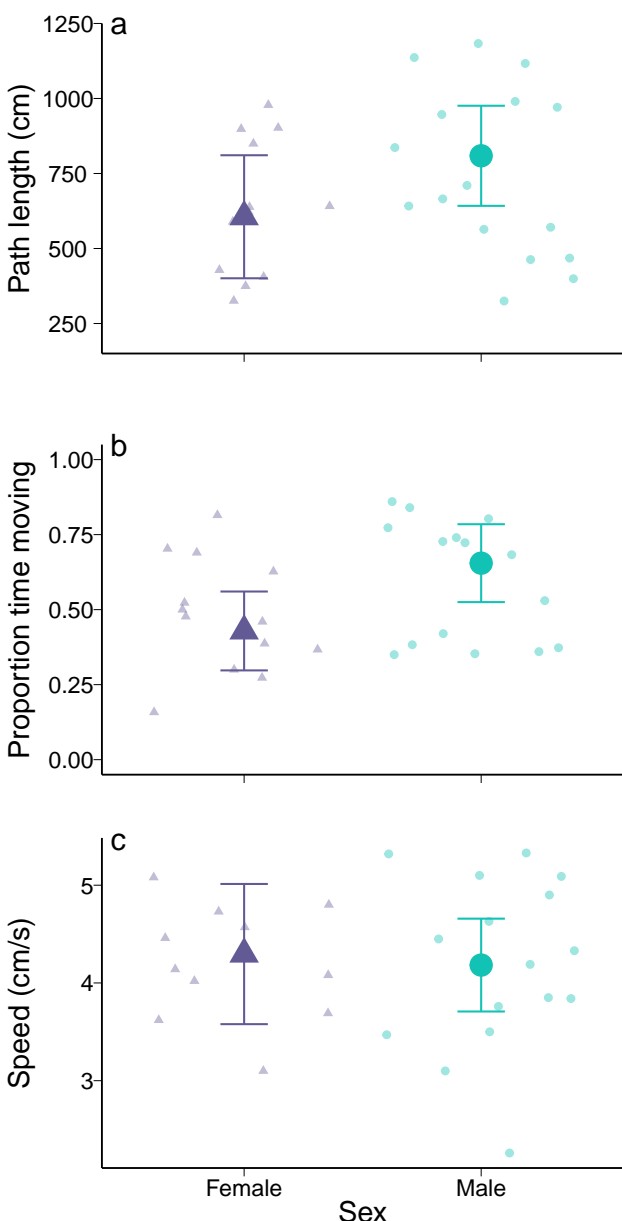

**Figure 5** **Exploratory behaviour of male and female European green crabs (*Carcinus maenas*). (A) Path length (cm), (B) proportion of time spent moving, and (C) speed (distance travelled/time spent moving) over a 5-min period.** $N = 16$ for males and N is variable for females ($n = 12$ for path length, $n = 13$ for proportion of time spent moving, $n = 11$ for average speed). Large points and error bars show model-estimated means and 95% confidence intervals at the mean carapace width of a crab in the experimental trials (*i.e.,* 66 mm). Observed data are shown as smaller points.

smaller claws on average, hence they consumed a smaller proportion of clams, on average. Our results suggest that trying to evaluate the potential impact of European green crabs on clam populations by sampling only males in functional response experiments could

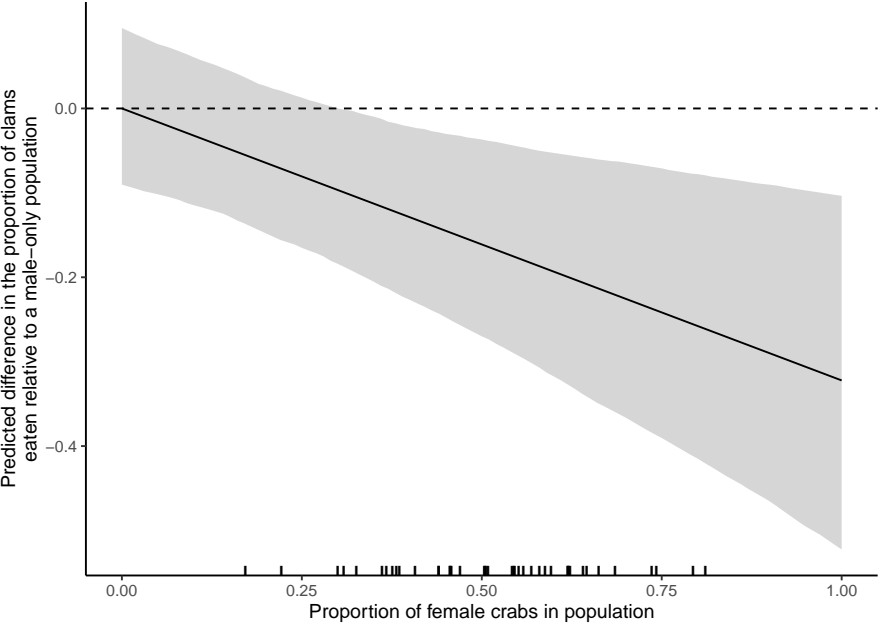

**Figure 6  The difference in the proportion of varnish clams (*Nuttallia obscurata*) expected to be eaten by European green crab (*Carcinus maenas*) populations of varying sex ratios, relative to a male-only population.** The black line shows the mean estimated proportional difference while the grey ribbon shows the bootstrapped 95% confidence interval. Observed sex ratios from the South Coast European Green Crab Control Project from repeated monthly sampling of four populations on the west coast of Vancouver Island are shown as ticks along the *x*-axis ($n = 42$).

often result in overestimates, especially when prey consumption is mediated by sexual dimorphism.

We found slight but non-significant differences in both attack rates and handling times between male and female green crabs. Attack rates represent the first two steps of the predation process: detection and attack (*Hassell, 1978*; *Lima, 2002*). A higher attack rate can be the result of heightened activity, which can lead to a higher probability of encounters with prey (*Huey & Pianka, 1981*; *Sweeney et al., 2013*; *DeLong, Uiterwaal & Dell, 2021*). For example, males might be expected to explore their surroundings more actively than females as they search for mates or establish competitive dominance, leading to higher rates of foraging (*Robles, Sweetnam & Dittman, 1989*; *Hunter & Naylor, 1993*). However, we could not detect differences in several metrics related to exploratory behaviour between male and female green crabs. In turn, handling time represents the last two steps of the predation process (capture and consumption) (*Hassell, 1978*; *Lima, 2002*). Sex-specific differences in handling times could stem from sex differences in prey choice (*Cordone et al., 2022*). For example, females could prefer smaller or different prey that are more profitable given their ability to access and consume these prey (*Weissburg, 1993*). Although female green crabs in our experiments generally had smaller crusher claws for their body size than males, they selected clams that were close to the average size of clams offered, as did males, which had proportionately larger crusher claws. We were therefore unable to identify specific

traits that might be associated with differences in functional metric components (see also *DeLong, Uiterwaal & Dell, 2021* for similar results with spiders).

The small, non-significant sex differences in attack rates and handling times observed nevertheless translated into a significant difference in the functional response ratios of males and females. The FRR of male crabs was double that of females. The FRR has been suggested as a reliable metric to predict the ecological impact of invasive species, regardless of context, with higher values reflecting higher potential impact (*Cuthbert et al., 2019*). If functional response ratios are consistent with impacts in the field, then male green crabs should be more damaging invaders than conspecific females, at least for clam populations.

The greater overall consumption capacity of male green crabs appears to be linked to their larger crusher claws. In several taxa, female predators often outperform their male counterparts, possibly due to their heavy investment in egg/offspring production (*e.g.*, *Donnelly & Phillips, 2001*). In many species of decapods, however, sexual selection has favoured the development of larger-sized crusher claws in males than in females (*e.g.*, *Lee, 1995*; *Pinn, Atkinson & Rogerson, 2001*; *Baeza & Asorey, 2012*), with concomitant consequences for foraging. In our experimental sample, where we selected green crabs within a limited range of carapace widths, males had crusher claws that were 26% larger than those of females. In a larger, random population sample, with a wider range of carapace sizes for both sexes, sexual dimorphism in crusher claw size was even greater (45%) (see also *Elner, 1980*; *Spooner, Coleman & Attrill, 2007* for similar results in the native range). The net effect of crusher claw size differences, given the relationships between clam consumption and crusher claw size uncovered here, is that females, on average, consumed a smaller proportion of clams than males (see also *Elner, 1980*). Note that prey consumption patterns could differ in moulting crabs and egg-bearing females (*Baeta et al., 2006*).

Based on our results, our modelling of consumption efficiency indicates that overestimation arising from the study of only males becomes nearly certain when at least 30% of individuals in a population are female. This means that overestimation of impact could occur even in populations with a male-biased sex ratio. Given the much smaller claw sizes of juveniles compared to adults (*Juanes et al., 2008*), a similar, or even greater, overestimation might occur when considering only adults rather than representative ratios of adults and juveniles in functional response experiments. Many have cautioned against extrapolating the results of functional responses to the wild. The unnatural simplicity of functional response experiments (*e.g.*, use of starved consumers, lack of competition or prey choice for consumers, lack of shelter for prey, etc.) can yield consumption rates that are unrealistically high (reviewed by *Griffen, 2021*). Nevertheless, differences in consumption metrics derived from functional response experiments appear to correlate qualitatively with differences in ecological impacts in the wild (*e.g.*, *Dick et al., 2013*; *Laverty et al., 2017*; *Howard et al., 2018*). Thus, while our simulation model might not reflect quantitatively the relative predatory impacts of male and female green crabs foraging on clams, we believe that it captures at least a qualitative difference. This difference, along with documented sex-related dietary differences (*Elner, 1980*; *Cordone et al., 2022*), suggests that sex might be an important characteristic to consider when using functional response experiments

to forecast the impacts of invasive green crabs, and perhaps of other novel invaders with marked sexual dimorphism that can affect foraging.

## CONCLUSIONS

We compared the functional responses exhibited by female and male invasive green crabs to test whether similar experiments that include only males are representative of this species' potential for impact. The functional response ratio, which integrates variation in attack rates and handling times and is thought to relate to the predatory impact of species in the wild, was twice as high for males than for females. While there was no difference in the proportion of clams consumed between males and females with similar claw sizes, females have smaller claws so they consumed a smaller proportion of clams. Taken together, our results suggest that the potential impact of European green crabs on clam populations could be overestimated for most populations if only males are considered in functional response experiments. We suggest that this phenomenon might be a general one, especially in species in which sexual dimorphism impacts foraging behaviour.

## ACKNOWLEDGEMENTS

Thank you to Claire Attridge, Andrew Bickell, and Kieran Cox for helping with varnish clam collection. Alex King and Crysta Stubbs from the Coastal Restoration Society allowed us to use and measure some of the green crabs collected within the traditional territorial waters of the Ahousaht and Tla-o-qui-aht First Nations as part of the South Coast European Green Crab Control Project—a collaborative effort between the Coastal Restoration Society, the Ahousaht First Nation (Maaqutusiis Hahoulthee Stewardship Society), the Tla-o-qui-aht First Nation, the T'Sou-ke First Nation and the Department of Fisheries and Oceans Canada. Site-specific data on green crab population sex ratio from trapping sites within Clayoquot Sound belong to the Ahousaht and Tla-o-qui-aht Nations and were shared with us for the purpose of supplementing this research project.

### Funding

This work was supported by the Natural Sciences and Engineering Research Council of Canada (No. RGPIN/03933-2017 to IMC). The South Coast European Green Crab Control Project is funded by the British Columbia Salmon Restoration and Innovation Fund. The funders had no role in study design, data collection and analysis, decision to publish, or preparation of the manuscript.

### Grant Disclosures

The following grant information was disclosed by the authors:
The Natural Sciences and Engineering Research Council of Canada: RGPIN/03933-2017.
The South Coast European Green Crab Control Project is funded by the British Columbia Salmon Restoration and Innovation Fund.

## Competing Interests

The authors declare there are no competing interests.

## Author Contributions

- Kiara R. Kattler conceived and designed the experiments, performed the experiments, analyzed the data, prepared figures and/or tables, authored or reviewed drafts of the article, and approved the final draft.
- Elizabeth M. Oishi conceived and designed the experiments, performed the experiments, authored or reviewed drafts of the article, and approved the final draft.
- Em G. Lim conceived and designed the experiments, analyzed the data, authored or reviewed drafts of the article, and approved the final draft.
- Hannah V. Watkins conceived and designed the experiments, analyzed the data, prepared figures and/or tables, authored or reviewed drafts of the article, and approved the final draft.
- Isabelle M. Côté conceived and designed the experiments, authored or reviewed drafts of the article, and approved the final draft.

## Data Availability

The data and code are available at Zenodo: Kiara Kattler. (2023). kiarakattler/green-crab-functional-response: Version 1 (v1.0). Zenodo. https://doi.org/10.5281/zenodo.7535182.

## Supplemental Information

Supplemental information for this article can be found online at http://dx.doi.org/10.7717/peerj.15424#supplemental-information.

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
