# Peer review of "Functional responses of male and female European green crabs suggest potential sex-specific impacts of invasion"

_PeerJ, doi:10.7717/peerj.15424_

## Round 0.1 · original submission · Major Revisions

I agree with the reviewers that the manuscript is, in general, well-written. However, there are some concerns raised on the validation of results (and their significance). Hence, I welcome a point-to-point rebuttal from the authors to address these concerns and suggestions.

·

Basic reporting

The paper is well-written in English with only a few minor grammatical edits:
Lines 48-49 -- “…and cause billions of dollars annually to coastal communities…” perhaps should be “…and cause the loss of billions…”
Lines 62-63 -- sentence subject noun “effect” is singular so requires the singular verb “has” (or change subject noun to “effects … have”
Line 99 – “Carcinus” need not be spelled out here because it has already been identified in line 75. “C. maenas” is now sufficient and appropriate.
Lines 120 & 134 – the word “haphazardly” is not the best / replace with “randomly”
Line 125 – insert “was” after “each” – “…each was supplied…”
Line 150—delete “a” after “…measured as …” – “...measured as part of …”
Line 185 – “…are counted…” should be “…were counted…”
Line 394 – “dimorphic” (adjective) should be “dimorphism” (noun) “… impacts foraging behaviour.”

All citations are listed in the “References” and visa-versa. References seem to be appropriate and sufficient with one exception:
Lines 75-84 – a 2020 review by Young & Elliott (Fishes, 5 “Life history and population dynamics of green crabs” summarizes much of this information and provides some more recent data so should be included here.
Lines 345-346 – why are we asked to “(see also Delong, Uiterwaal & Dell, 2021)”? Why? For what? Clarification is needed.
Lines 436-439 – I’m not clear about what this reference is. Should something be italicized?
Line 560 —“Behrens Yamada, S.” should be listed alphabetically after the “Bax” reference. That’s how Sylvia does it in her own papers.

No figure legends are provided so Figures 1-6 cannot be properly evaluated. However, if they are printed in black & white (as I and many readers do) it is almost impossible to distinguish the male and female dots in figure 3. I suggest using two distinguishable symbols rather than just different colored dots. Similarly, the shaded areas in other figures are difficult to interpret in black & white.
The supplemental figures (S1-S3) include legends but my comments re dot colors also apply to Figure S2.
In Table S1 column headings “Lower Bca” and “Upper Bca” are not identified in the legend.

The manuscript is a “self-contained” study and not a subdivision of a larger project.

Experimental design

The methods are appropriate for the research being conducted. I have a few specific questions that require clarification:
Line 136 – were the number and size of clams consumed recorded in addition to clams remaining?
Line 150 – if “Crabs were euthanized immediately after they were captured …” then how could they be used in experiments (Lines 156-160)? Clarification of methods needed here.
Line 183 – “trapping at a site / each site…”?
Line 184 – “should “between” be “among”?

Validity of the findings

The question being investigated, if predation differences exist between male and female crabs, will be of interest to others working in this area. The finding that there are differences could, as the authors suggest, lead to inaccurate estimates of the impact of these crabs on prey (clam) populations, and potentially could apply to other predator species as well. The conclusions are supported by the findings.

Results:
Line 251 –explain what “Type II functional response” means

Reviewer 2 ·

Basic reporting

The manuscript is well written and the subject is of great interest.
The use of English is professional and within academic standards.
Figures and tables are well defined and helps the reader to understand better the findings of the current study.

Introduction
Lines 74-75: there are two native Carcinus in the Mediterranean region; the other is Carcinus aestuarii Nardo, 1847

Experimental design

The article is suitable for the journal, and falls within its scopes
The research questions are clearly presented and meaningful.
Some minor cooments:

Line 123: Introduction of? please specify
Lines 123-124: please add further information on the range of (a) grabs’ width range and (b) on claws’ width.
Line 127: it seems that the experimental area was not fully isolated by the rest of the lab facilities, please discuss this and the potential biases introduced at the discussion section.
Line 137: please add further information on the protocol used for euthanizing the specimens. It would help the readers to understand that all ethical procedures were followed. Probably, it would something really simple...
Lines 138-139: Refer the house and the model of each instrument.

Validity of the findings

Experiments like the one presented here can be useful in our understanding of invasion biology. Especially in cases, with “global” invaders such as the green crab.
Each experiment was replicated many times, allowing obtaining a relatively good data set, though that someone might argue that these experiments are not very representative for the “actual” status in nature. However, they are necessary to build up knowledge starting from the basics. Too complex experimental designs can lead to vague assumptions.

Additional comments

Discussion
An interesting subject is the fact that as mentioned by the authors green crabs are native to European-Mediterranean waters; one of the most notorious invaders there is the blue crab Callinectes sapidus. The species is posing serious threats in ecology and economy.
Kampouris et al (2019) is among the few studies, dealing with the foraging ecology of the species. The study found that differences in foraging behavior could be affected by the sex of the animals and the size. I would appreciate a small paragraph discussing the invasion (in terms of feeding preferences) of the green crab in the Americas and the invasion of the blue crab in Europe.
Kampouris, T.E. Porter, J.S., and Sanderson, W.G. 2019. Callinectes sapidus Rathbun, 1896 (Brachyura: Portunidae): An assessment on its diet and foraging behaviour, Thermaikos Gulf, NW Aegean Sea, Greece: Evidence for ecological and economic impacts

Reviewer 3 ·

Basic reporting

As a non native English speaker I don't feel confident to evaluate the writing. I found English to be clear and technically correct.

I found enough background in the field of functional response and per capita effects. Authors take into account recent reviews in its field. However, I have been missing a greater depth in the field of crab diet. Carcinus maenas is one of the most studied crabs in the world and there is a wide variety of literature on its diet. There is relevant literature that describes differences between the diets of males and females and between juveniles and adults. This aspect is relevant, since the differences between diets, the role of gender differences in trophic positions and the occurrence of prey items between sexes or ontogenic groups is well described in the literature. I’m wondering if there could exist a more complex scenario than the difference between sexes with respect to a specific prey item. I miss a deeper dimension of this aspect. In the present study impact is related to prey consumption. From this, a major effort describing heterogeneity in Carcinus maenas diet is required. and could enrich the discussion and at some point tone down the conclusions of the present study.

Experimental design

Methods were generally described with sufficient information to be reproducible. The experiment is well designed and incorporates high technical standards. Authors explore all possible dimensions of foraging and statistic are sound. Despite this, there are some things that should be explained in greater detail or clarified:

1.- The use of a non-indigenous prey to mesure impacts in natural habitats should be justfied

2.- It could be relevant to include the time (hour) were it was recorded the "Exploratory behaviour" and if it was the same for the different groups. Activity in decapods are related to cyrcadian rithms. Time of teh day is relevant when measuring activity. I recoment a littel explanation on this.

3.- Sex-ration in natural populations: The total capture obtained from deppletiosn should be used to calculate the sex ratio. The accumulated catch (5 days) is representative of the population tha existed in the area. The single intake of the first day to calculate the sex-artio should result in a higher proportion of males. This is an artifice because the sampler (bait traps) is passive and favors the initial capture of the larger and dominant specimens. In the case of crabs, the males. I invite the authors, if possible, to calculate the sex ratios based on the accumulated catch. This should shed a greater proportion of females and juveniles than of adult males.

Validity of the findings

My major concern is the main statment of the study. I didn't find true differences in the foraging behaviour of males and females. Gender dimension around diet could exist as resource partitioning or different trophic position but this is not causality of impact.

How do the authors justify that adult males have a higher rate of prey consumption or higher energy requirements than females? This aspect is relevant and requires an in-depth discussion, since it contradicts the general trend.

I urge authors to avoid expressions like "slight" "small" differences. The differences must be significant or not significant. I suggest the authors base the discussion on their results. Avoid reference to slight differences or suggestions. It would be good to include or reflect on the limitations and uncertainty that exist when working with a specific prey in a heterogeneous and diverse diet. I see value in this work and a lot of effort.

---

## Round 0.2 · Minor Revisions

The revised version of the manuscript is almost ready to be accepted, pending some minor concerns from reviewer 3. I do agree with Reviewer 3 that while gender might play a role, factors such as maturation status, diet preference, and habitat preference might be more influential. A discussion on this would be splendid.

·

Basic reporting

standards met / my comments from earlier review have been addressed satisfactorily

Experimental design

standards met / no comment

Validity of the findings

standards met / no comment

Additional comments

All of my comments from the earlier review have been addressed and I am satisfied. This revision is an improved version of the article.

Reviewer 3 ·

Basic reporting

GENERAL Response 2

Dear Editor,

The authors have responded to most of my concerns. Some suggestions have not been accepted but have been argued and justified.

As requested by the authors I have provided some references on the diet of Carcinus maenas, I also suggest exploring other sister species such as C. estuaris or other portunids. However, this is a recommendation and not an imposition.

I have tried to express my concern about the effect that differences between juveniles and adults may have on the study, leaving aside the gender of the crabs. I understand that this is not the aim of the study and therefore leave it to the authors to reflect on this fact.

In my view the work has improved in the other aspects. It only remains to congratulate the authors for this work.

Best regards.

R1 is the initial question
R2 is the answer to the authors
* * *
Reviewer 3
Basic reporting
As a non-native English speaker, I don't feel confident to evaluate the writing. I found English to be clear and technically correct.

R1 REVIEWER 3: I found enough background in the field of functional response and per capita effects. Authors consider recent reviews in its field. However, I have been missing a greater depth in the field of crab diet. Carcinus maenas is one of the most studied crabs in the world and there is a wide variety of literature on its diet. There is relevant literature that describes differences between the diets of males and females and between juveniles and adults. This aspect is relevant, since the differences between diets, the role of gender differences in trophic positions and the occurrence of prey items between sexes or ontogenic groups is well described in the literature. I’m wondering if there could exist a more complex scenario than the difference between sexes with respect to a specific prey item. I miss a deeper dimension of this aspect. In the present study impact is related to prey consumption. From this, a major effort describing heterogeneity in Carcinus maenas diet is required. and could enrich the discussion and at some point, tone down the conclusions of the present study.
RESPONSE: Extensive descriptions of green crab diet (beyond what we added) and of ontogenetic diet changes have been reviewed by others (which we cite, e.g. Klassen and Locke 2007) so we do not feel it necessary to duplicate these efforts here.
* * *
R2 Reviewer 3: Accepted

Authors RESPONCE: However, we have added that European green crabs on the Pacific coast of North America are generalist predators that consume predominantly mollusks and crustaceans (lines 109-111). Limiting the description of the diet to this part of the invaded range (and more specifically British Columbia) seems appropriate for putting our results in context and justifying our choice of experimental prey. We would be grateful if the reviewer could point to specific studies that document sex differences in invasive green crab diet. We cited the one we could find (Cordone et al.) but are not aware of others.

R2 REVIEWER 3: For sure, here are some examples (no time consuming to make the search). The suggestions are constructive and with the intention to enrich the present document. At the same time, it is not a mandatory. The inclusion of these articles is just a suggestion.

Elner, R.W., 1980. The influence of temperature, sex and chela size in the foraging strategy of the shore crab, Carcinus maenas (L.). Marine & Freshwater Behaviour & Phy, 7(1), pp.15-24.

Spooner, E.H., Coleman, R.A. and Attrill, M.J., 2007. Sex differences in body morphology and multitrophic interactions involving the foraging behaviour of the crab Carcinus maenas. Marine Ecology, 28(3), pp.394-403.

Pardal, M., Baeta, A., Marques, J. and Cabral, H., 2006. Feeding ecology of the green crab, Carcinus maenas (L., 1758) in a temperate estuary, Portugal. Crustaceana, 79(10), pp.1181-1193.

Conspecifics, pleas take in to account that Carcinus maenas and Carcinus estuary can hybridize. So, diet studies (even not the same species) should be taken into a count. In another degree. Other studies around Portunidae crabs to find sex and ontogenic difference it ‘will be useful.
* * *
Autghors RESPONSE: We now make it clear that our study focuses on a single prey species and that results may be different for other prey species whose consumption is not, or is less, affected by crab sexual dimorphism.

R2 REVIEWER 3: Accepted

Experimental design

R1 REVIEWER 3: Methods were generally described with sufficient information to be reproducible. The experiment is well designed and incorporates high technical standards. Authors explore all possible dimensions of foraging and statistic are sound. Despite this, there are some things that should be explained in greater detail or clarified:

R1 REVIEWER 3: 1.- The use of a non-indigenous prey to measure impacts in natural habitats should be justified

Authors RESPONSE: Although the prey species chosen, varnish clams, is not native to BC, it is among the most abundant bivalves now encountered in many parts of the province, its habitat distribution overlaps with that of green crabs, and it is a preferred prey for local green crabs. We therefore expect a high likelihood of encounters between green crabs and this non-native prey species. We have added a stronger justification for using varnish clam as prey species (lines 111-115).

R2 REVIEWER 3: Accepted


R1 REVIEWER 3: 2.- It could be relevant to include the time (hour) where it was recorded the "Exploratory behaviour" and if it was the same for the different groups. Activity in decapods are related to circadian rhythms. Time of the day is relevant when measuring activity. I recommend a little explanation on this.

Authors RESPONSE: We have added the time of day when the exploratory behaviour experiments were run (08:00 to noon), which overlaps with the timing of the functional response experiments (line 178). We also added the time of the functional response experiments (line 148).

R2 REVIEWER 3: Accepted


R1 REVIEWER 3: 3.- Sex-ratio in natural populations: The total capture obtained from depletion should be used to calculate the sex ratio. The accumulated catch (5 days) is representative of the population that existed in the area. The single intake of the first day to calculate the sex-ratio should result in a higher proportion of males. This is an artifice because the sampler (bait traps) is passive and favors the initial capture of the larger and dominant specimens. In the case of crabs, the males. I invite the authors, if possible, to calculate the sex ratios based on the accumulated catch. This should shed a greater proportion of females and juveniles than of adult males.

Authors RESPONSE: The reviewer is absolutely correct, and we in fact wrongly described what we did in the original manuscript. In the original MS, the “first day catch” sex ratios were actually only used to generate the ticks on Figure 6, not to calculate the site-specific ratios. We have corrected the description of the analysis (now reading as “the cumulative sex ratio of each trapping period at a site was used”, lines 202-204) and have replaced the ticks in Figure 6 with the accumulated catch. The ratios presented for each site (in the Results and in Figure S3) are based on all available data (accumulated catch). The individual data points in S3 are the ratios for each day (i.e., not accumulated over the sampling period), but since the logistic regression used to estimate the site-specific ratios was weighted by the number of crabs caught each day, the results are identical whether you sum the number of crabs caught over each period into a single observation or calculate a daily proportion. The results regarding the percentage of sites with over 31% female crabs has now been updated (line 347-348).

R2 REVIEWER 3: Accepted

Validity of the findings

R1 REVIEWER 3: My major concern is the main statement of the study. I didn't find true differences in the foraging behaviour of males and females. Gender dimension around diet could exist as resource partitioning or different trophic position but this is not causality of impact.

Authors RESPONSE: Please refer to the comment below about significant differences where we elaborate on our conclusions. We now emphasize the functional response ratio results (lines 31, 224-232, 292-295, 355-357, 381-383, 422-424) to support our conclusions about the sex differences we observed in the study. This index presents an integrated picture of the small differences found in the individual functional response metrics.

We are unsure as to what the reviewer means regarding the causality of impact. We agree that there is potential for male and female green crabs to be occupying slightly different trophic positions and that there may be resource partitioning between sexes, but a more obvious reason for any differences (especially when dealing with a single prey species consumed by crushing) is the sex difference in claw size of crabs (lines 358-360).

R2 REVIEWER 3: I understand that I have failed to adequately convey my concern to the authors. The authors base their work on the sex ratio. On the basis of this ratio, where adults and juveniles are mixed, the requirements of males and females have been compared. They conclude that, as males have a higher prey "consumption" and are dominant (in this study) in the population, the impact of the species may be underestimated. What I am trying to express is that the difference between males and females may be smaller than the difference between juveniles and adults. If it is related to chela size, there are effects such as claw loss or other events that may interfere. Therefore, gender may be a factor, but I wanted the authors to reflect on whether these differences are greater, for example between adults and juveniles, regardless of sex. I hope the authors can reflect on this. Again, this is a constructive comment.

e.g., Mathews, L.M., McKnight, A.E., Avery, R. and Lee, K.T., 1999. Incidence of autotomy in New England populations of green crabs, Carcinus maenas, and an examination of the effect of claw autotomy on diet. Journal of Crustacean Biology, 19(4), pp.713-719.


R1 REVIEWER 3: How do the authors justify that adult males have a higher rate of prey consumption or higher energy requirements than females? This aspect is relevant and requires an in-depth discussion, since it contradicts the general trend.

Author RESPONSE: It is true that females might have higher energy/prey consumption requirements during egg production and incubation, but we had no ovigerous females in our experiment and females were smaller than males. We therefore attributed the greater consumption capacity of male green crabs to their larger body size and, more specifically, to their larger crusher claw size (lines 358-360). However, we have added a sentence highlighting the potential for a different pattern of prey consumption in breeding females (lines 399-400).
REVIEWER 3: Accepted

---

## Round 0.3 · accepted · Accept

Thank you authors for following through with the reviewer's comments. The manuscript is now clearer and ready for publication.

Reviewer 3 ·

Basic reporting

no comment

Experimental design

no comment

Validity of the findings

no comment

Additional comments

no comment